# LGBTIQ CALD People’s Experiences of Intimate Partner Violence: A Systematic Literature Review

**DOI:** 10.3390/ijerph192315843

**Published:** 2022-11-28

**Authors:** Alex Workman, Erin Kruger, Sowbhagya Micheal, Tinashe Dune

**Affiliations:** 1School of Health Science, Western Sydney University, Locked Bag 1797, Penrith 2751, Australia; 2School of Social Science, Western Sydney University, Locked Bag 1797, Penrith 2751, Australia; 3School of Medicine, Western Sydney University, Locked Bag 1797, Penrith 2751, Australia; 4Translational Health Research Institute, Western Sydney University, Locked Bag 1797, Penrith 2751, Australia

**Keywords:** intimate partner violence, sexuality, CALD, resilience, survival, violence, LGBTIQ

## Abstract

Background: Experience of surviving intimate partner violence (IPV) is well documented in research, policing practices, newspapers, and awareness campaigns domestically and internationally. Arguably, those who have survived IPV and have their experiences reflected within society undergo a transformative experience of empowerment. As society recognises and validates their experience has occurred, and responds to it, accordingly, as some survivors have targeted services and interventions to assist in this transformation. However, for LGBTIQ-identifying peoples, experiences of IPV are poorly understood in contemporary society, which is further exacerbated for LGBTIQ-identifying CALD people as they continue to remain hidden. Aim and objective: The systematic literature review aims to explore the experiences of this group in their development of resilience following an abusive and violent relationship. Methods: Of the potential 230 identified studies, 5 studies met the eligibility criteria. In line with the eligibility criteria, these studies were first reviewed by title, then by abstract and then by full text. Of those studies which the research team deemed relevant for inclusion, their reference lists where also reviewed to determine if any further relevant studies could be identified using this strategy. As a result of the above process, five (5) studies met the eligibility criteria and were included in the study. Results: From data extraction, three major themes emerged: *Intimate Partner Violence as Experienced by LGBTIQ survivors, Marginalised Identity and Types of Survivorship*. While refined, these themes capture a more robust set of sub-themes that identify the diverse ways in which LGBTIQ survivors experience responses to their victimised status as experienced in IPV situations. Conclusions: Overall, the review found that resilient outcomes for LGBTIQ CALD survivors remain misunderstood and missing within the literature. There is a propensity to focus on negative coping strategies and an exclusive focus on LGBTIQ CALD vulnerabilities. Future research requires investigation into strategies and support that move beyond coping to include resilient outcomes and support systems that allow manifestations of resilience.

## 1. Introduction

Intimate partner violence (IPV) occurs when an individual in an intimate relationship coerces, controls, demeans or abuses their partner [1]. Falling under domestic violence (DV), IPV can be physical, psychological, emotional, spiritual, geographical, and economical. Victims may experience one or a multitude of these in isolated periods, concurrently or for extended periods [1]. It is important to note that some forms of IPV are unique to lesbian, gay, bisexual, transgender, intersex and queer or questioning (LGBTIQ) populations. LGBTIQ IPV may include withholding HIV medication or knowingly exposing their partner to HIV, outing a person, homophobia, biphobia, transphobia and heterosexism [2,3]. These distinctive types of abuse are unique and challenging to identify and therefore require targeted interventions to address.

While DV is the commonly accepted umbrella term, some additional characteristics distinguish DV from IPV. In DV, the victim and perpetrator do not need to be intimate partners. Further, DV perpetrators can be between family, including children, cousins, uncles, aunts, grandparents, and parents or between friends or housemates who live under the same roof [4]. Commonly, DV involves male dominance and female subjugation [5,6]. Family Violence (FV) also falls under DV and is similarly characterised by the exertion of control, dominance, and power over another person within a family unit/relationship [7]. Bates and Taylor [1] found that DV and FV do not account for LGBTIQ-identifying experiences of IPV—a gap which this systematic review seeks to explore.

Current policy, advocacy and media reporting within Australia focuses on DV and FV [3]. These reports centre on White, cisgender, heterosexual women’s victimised experiences that ultimately shape contemporary societal understandings of violence. Further, many groups remain overlooked, notably the LGBTIQ community [3,8,9,10]. Qualitative understandings of victims’ experiences are often missing in the current policy, advocacy, and media reporting from the LGBTIQ perspective. As a result, this group’s experiences are underreported, undetected, and unnoticed by formal institutions. There are multiple aspects to a person’s identity such as race, ethnicity, gender, sex, sexuality, disability status, religion, social class, education, health, and geographic location [11]. Crenshaw [12] terms this intersectionality, which critiques formal institutional outcomes in light of the complexities of experience and identity [13]. Social justice outcomes are heavily influenced by intersectionality, with some identities experiencing less favourable outcomes than others [14]. For example, a Black, transgender woman with a disability and low education level may have different opportunities and outcomes from a Black woman with a disability and low education level [15,16]. Notably, people who have marginalised identities are more likely to have poorer health, wellbeing and criminal justice outcomes as they are less represented in policy, legislation and across all forms of media [3,8]. As such, the nexus of LGBTIQ- and culturally and linguistically diverse (CALD)-identity requires investigation given the compounded challenges this group faces. In Australia, CALD is a term that best describes the experiences of those from ethnically, racially, and religiously diverse communities. However, this excludes people who are Aboriginal and Torres Strait Islander and disabled people, such as those who use sign language [17]. The definition of CALD has conflicting meanings, despite its widespread use, eligibility depends on ones country of birth, language spoken at home besides English and depends on one’s culture or ethnicity. For some first-generation migrants from the UK could be considered CALD. However, in Australia, people who are Anglo-Saxon, Anglo-Celtic or Aboriginal or Torres Strait Islander are excluded [17]. As the CALD definition places emphasis on linguistic diversity, but excludes disability, emphasis is placed on ethnic and racial differences. 

For LGBTIQ CALD IPV survivors, there are many concurrent prejudices and societal barriers which may deter this diverse group from engaging with police officers, shelters and other DV services [18,19]. Formby [19] found that LGBTIQ CALD people, particularly those of Asian and Black identity, face significant cultural barriers. Some barriers manifest from racism, homophobia, transphobia, and biphobia that reduce opportunities for acceptance from their immediate social supports and local institutions. For example, Miles-Johnson and Wang [20] found that in Beijing, *Filial Piety* is applied to ensure LGB people remain in the closet as it brings dishonour to their social network. LGBTIQ CALD survivors are in a precarious position based on the intersections of their identities. Moreover, while there are cultural barriers for LGBTIQ CALD survivors, there is also the significant issue of systemic institutional racism. Some victims may let their experience be known but face discrimination from institutions such as the criminal justice system, health services and non-governmental organisations, to name a few, due to their CALD identity [21].

### 1.1. LGBTIQ CALD IPV Survivors’ Experiences of Resilience

Resilience is the human ability to adapt in the face of hardship, trauma, adversity, violence and other life stressors [22]. The process of being and becoming resilient is highly individualised. It is based upon a multitude of intersectional factors (e.g., sex, gender, age, race, ethnicity, sexuality status, gendered identity, disability status, Indigenous status) [12] and social determinants of health [23](e.g., socioeconomic status, geographical location, education, work, income). Donovan and Barnes [18] found that there are limited institutional social supports for LGBTIQ CALD survivors of IPV due to homophobia, biphobia, transphobia and racism These attitudes coalesce to create a unique set of experiences that, according to Donovan and Barnes [18], has not been accurately captured in qualitative research. While the definition and societal understandings of resilience evolve through time, Newman [22] identifies the importance of understanding resilience as multidimensional and complex. According to Friborg and colleagues [24], the principles of resilience include:When a person is given a sense of purpose;Belief in one’s abilities;Developing strong social networks;Embracing change;Being optimistic;Nurturing oneself;Developing problem-solving skills;Establishing goals, taking action, and keep working on their skills.

Individuals all have the capabilities to become resilient or strengthen their resilience; arguably, some people may require additional support, such as those who experience(d) IPV [3]. It is, therefore, crucial to understand the experiences and factors that help or hinder this group in their manifestations or development of resilience following an abusive and violent relationship.

### 1.2. Theoretical Framework

To operationalise the impact of identity, interpersonal connections, institutional influences and society, this systematic review uses Bronfenbrenner’s [25] socioecological theory. Socioecological theory acknowledges that a person’s development is influenced by their socialisation constructed by forces outside of their control. According to Bronfenbrenner [25], there are five interconnected yet interconnected levels that influence us all (see Figure 1).

This theory’s premise is that individuals develop within a set of spheres, much like a Russian nesting doll (see Figure 2), by external forces that heavily influence LGBTIQ individuals’ development outside of their control. These external forces directly reshape the type of victimisation an LGBTIQ person would experience, including their rates of disclosure, intervention, services and awareness of LGBTIQ IPV. Contrariwise, heterosexual White women’s experiences are more readily documented within research, policy, legislation, advocacy, and awareness campaigns [3]. However, some individuals’ survivorship experiences are negatively impacted due to the social stigmas attached to one or more of their identities, such as being black, identifying as a lesbian and being transgender [26]. Due to these compounding factors, these individuals are more likely to experience social exclusion, subjugation, marginalisation, and repeat victimisation [27]. This minority status brings its own sets of challenges such as underreporting, limited policy development, little to no advocacy, and experiencing more significant levels of dehumanisation as a result of their social invisibility.

The microsystem is the immediate environment, the mesosystem is the individuals’ social connections, the exosystem is their indirect environment, the macrosystem is the persons social and cultural values, and the chronosystem which are changes over a period of time [25]. While each system mutually reinforces the chronosystem, it does not directly play a role in the individuals’ lives; however, it directly influences the other systems. Therefore, this review will engage with the principles of socio-ecological theory (e.g., Bronfenbrenner, [25]). Doing so will allow for a multidimensional view of the interactions and relationships between a wide range of factors within a person’s environment. The socio-ecological theory supports this as it helps identify constructs, interactions, and experiences between individuals and various levels of their environment. It helps to provide additional and holistic insights into the social intricacies and dimensions of gender, sexuality and CALD identities are shaped by the individual and their environment, which other studies tend to overlook (e.g., see Strasser et al., [28]). Like any individual, the socio-ecological environment of an LGBTIQ CALD-identifying individual includes a complex network of structures that progressively shape (and is shaped by) the individual as they traverse through it [25]. Hence, this systematic review seeks to fill this gap by qualitatively capturing LGBTIQ CALD identifying survivors’ experiences and their manifestations of resilience.

### 1.3. Research Questions

To inform the aim of this review, this systematic review sought to collate evidence to answer the following research questions:How do LGBTIQ people experience survivorship and manifestations of resilience, as discussed in the peer-reviewed literature?How are experiences of survivorship reported on within studies concerning marginalised LGBTIQ people?How are experiences of coping and vulnerability as precursors for survivorship and understandings of resilience for marginalised LGBTIQ people discussed within peer-reviewed literature?

## 2. Method

In line with Preferred Reporting Items for Systematic Reviews and Meta-Analyses (PRISMA) guidelines [29], a systematic review in line with the theoretical framework to understand the experiences and factors that help or hinder LGBTIQ in their manifestations or development of resilience following an abusive and violent relationship.

### 2.1. Search Strategy

The literature search included five electronic databases in psychology, health, and social sciences: EBSCOHOST, ProQuest Central, Taylor and Francis, CINAHL, and INFORMIT. The search for published peer-reviewed literature in English was undertaken between August 2014 and September 2020. Following Dune, Caputi and Walker [30], a step-by-step search strategy was employed (see Figure 2). A preliminary search of ProQuest Central was undertaken to identify the keywords contained in study titles and abstracts and ascertain index terms used to describe articles. Pertinent keywords were discussed, expanded, and refined with the primary supervisors. A second search, using all identified keywords, was conducted across the five databases indicated. Finally, the reference lists of all included studies were examined for additional literature. Details of the search strategy, including the search terms and combinations, are summarised in Table 1. Details of the included studies and their research methods including theory and participant data are summarised in Table 2.

### 2.2. Data Synthesis and Interpretation

The review analysed the literature using a thematic approach developed by Thomas and Harden [31] to extract, synthesise, analyse, and interpret the findings of the included literature. Three steps were followed: (1) line by line coding of the results, discussion, and conclusion sections of the primary studies; (2) development of descriptive themes; and (3) generation of analytical themes towards a synthesized presentation of results. The first author completed a preliminary synthesis of primary data followed by a review and disagreement resolution with the supervisory panel.

### 2.3. Study Quality

Study quality, as shown in Table 3, included three major assessment criteria: (a) validity, (b) results and (c) local relevance of results. There are 12 for quantitative cohort studies. Responses include (a) yes, (b) cannot tell or (c) no. A qualitative indication of quality based on the responses to the sub-questions provided an overall indication of a study’s quality level: strong (S), moderate (M) and weak (W). Studies which received a “yes” response to all questions were determined to be of strong quality. Studies that received three “can’t tell” or “no” to any questions were rated as moderate. Studies with more than four “can’t tell” or “no” were determined to be of weak quality.

## 3. Results

From the 230 potentially relevant articles identified, 5 articles were included in this systematic review (see Figure 2).

### 3.1. Research Foci and Theoretical Approach

The included studies primarily focused on the experiences of marginalised victims (five), intimate partner violence (IPV) (three), dating violence (one) and sexual violence (two). Out of the five studies included in this study, three studies focused exclusively on LGBTQ-identifying individuals, two studies identified LGBTIQ and culturally and linguistically diverse (CALD), and one study included religious minorities. Additionally, the type of survivorship discussed within the peer-reviewed literature focused exclusively on vulnerability and coping (five). All studies included participants (five), and three studies included the use of theories. In those instances, the theories were supporting survivors’ self-concept theory, attribution theory, planned behaviour theory, minority stress theory and intersectionality theory, the remaining studies (two) did not specify the use of any theory (see Table 2).

### 3.2. Research Design and Methodology

Only four studies indicated the use of a methodological framework, where the authors advised that the use of a methodology informed their data collection process within their article. Four studies used quantitative methodology, with the use of surveys cited as the most common data collection strategy. Given the emphasis on quantitative methods, a variety of statistical analyses were applied to this review, including chi-square *t*-tests (2), *t*-tests (1), latent class analysis (1), descriptive statistical analysis (1) and logistic regression (1).

## 4. Major Findings

Following line-by-line coding of the extracted results and discussion sections from each study, three major themes emerged including: *Intimate Partner Violence as Experienced by LGBTIQ Survivors, Marginalised Identity and Types of Survivorship.*

Theme 1 addresses research question 1, by reflecting on the nuances of survival for LGBTIQ people. Theme 2 addresses questions 2 and 3 by discussing the diverse forms of survivorship and how coping and vulnerabilities are precursors for resilient outcomes. Theme 3 address question 3 by discussing the types of survivorship are discussed in the selected literature. While refined, these themes capture a more robust set of sub-themes that identify the diverse ways in which LGBTIQ survivors experience responses to their victimised status as experienced in IPV situations. Nevertheless, they also identify the ways society responds to their experiences, prioritises different sexual and ethnic identities and how society may perpetuate repeat victimisation. In brevity and readability, the significant findings are accompanied only by a few example citations.

### 4.1. Intimate Partner Violence as Experienced by LGBTIQ Survivors

Few studies provided robust data concerning all LGBTIQ survivors of IPV ([28,32,33]. Strasser et al. [28] identify IPV for gay men as the third-largest public health issue, even though there is a lack of awareness and understanding of the experience of gay men and the strategies they adopt to survive IPV relationships (such as coping, vulnerability, resilience, or grit). Pittman et al. [33] mirror this lack of awareness of lesbian women. Specifically, lesbians in the process of coming to terms with their sexuality are more likely to experience IPV, although social stigmas associated with their identity remain hidden from data collection, awareness campaigns and social justice responses. This type of violence is known as “outing” and is a common form of power and control within LGBTIQ IPV relationships due to the stressors associated with coming out, including familial rejection [34].

Lou et al. [35] identify this as unique to LGBTIQ people. Nevertheless, there are added layers of complexity for individuals with multiple marginalised identities or that come from cultural backgrounds which do not outwardly accept non-cis-hetero identities [32]. There was a propensity to compound IPV experiences with sexual violence [32], which does not consider the definitional nuances of each type of violence. For example, IPV is commonly understood as the control processes that one person uses to demean, devalue, and dehumanise their victim [9]. In contrast, sexual violence can be used in IPV relationships; however, it may occur outside of relationships where the victim may not necessarily know their perpetrator and is done for humiliation and degradation [9]. These slight variances and lack of clarification convolute understandings of these forms of violence, which are often unique from one another and can arise as isolated experiences.

### 4.2. Marginalised LGBTIQ Identity as Experienced by Intimate Partner Violence Survivors

There are many forms of marginalised identity. These include having a disability, being a refugee, or a migrant, looking visibly different through racial or ethnic differences, having a gendered identity that does not conform to the rigid socially constructed identities such as male or female, sexuality, and faith not considered dominant. Individuals can belong to these marginalised and vulnerable identities in a combination of one or more. These multiple layers of identity are commonly known as intersections and are associated with intersectionality theory. However, when discussing issues such as IPV, the more marginalised parts of a person’s identity one has, the less likely they are to receive the social validity of their experiences. Pittman et al. [33] note that for lesbian Latina women, there are significant misunderstandings of what constitutes abuse. Emotional abuse is not considered IPV within this cultural group, and this can increase the hiddenness and underreported nature of the non-physical aspects of IPV.

Moreover, Strasser et al. [28] found that it is challenging to identify gay male experiences of IPV due to the significant social stigmas attached to homosexuality. However, physical and sexual violence (within the relationship) directed at this group is slowly gaining awareness and validation, albeit psychological and sexual coercion are still not discussed as extensively. Strasser et al. [28], who interviewed 100 participants, further identified that when an individual belongs to the non-White category for victimisation, they experience some form of IPV at a rate of 51.4%. This identification of victimhood proposes that the more vulnerable categories one belongs to, the more likely they are to experience IPV, which will go undetected, undisclosed, and unrecorded. 

### 4.3. Types of Survivorship as Experienced by LGBTIQ Survivors

Many forms of violence experienced by LGBTIQ people go underreported, and when the violence is reported, there is a focus on physical violence only [36]. Within the literature, the focus of survivorship—no matter what type of violence experienced—focused exclusively on coping strategies and vulnerabilities of being marginalised as it relates to one’s sexual identity (see Strasser et al. [28]; Luo et al. [35]; Whitton et al. [36]; Pittman et al. [33]; Edwards et al. [32]). For marginalised LGBTIQ survivors, surviving brings with it a unique set of challenges. Notably, these challenges stem from their inability to disclose the victimisation they experience when living in an IPV relationship [28] Arguably, this creates negative coping strategies where Strasser et al. [28] found that gay men are more likely to use illicit substances to cope with their victim status.

While social support can be positive, many LGBTIQ victims disclose their experience to informal social support. Informal support is commonly understood as the support an individual seeks based on their social networks. For example, individuals may seek the advice of a friend, a trusted family member or a colleague. In comparison, formal supports are institutions such as a doctor, counsellor, psychologist/psychiatrist, social worker, police officer, etc. Moreover, these negative coping strategies are further pathologised when these vulnerable groups disclose their experiences to friends or family members [35]. Where the recipient of the disclosure may not know their LGBTIQ status or not accept that part of their identity and reject it—in turn, this may lead to repeat victimisation. This rejection further solidifies why victims choose to stay with their abuser related to the disbelief and dismissal of their identity. For example, Luo et al. [35] (2012) found that for gay men, survival experiences decreased when the abuse was psychological, sexual, or they experienced coercion, which forced them to “come out”. Arguably, this contrasts physical abuse, which was found to intensify trauma in the IPV context. These effects of experiencing abuse manifest in diverse ways and depend on the survivor’s experiences; however, there is a dependency to rely on illicit substances and alcohol dependency to cope with the experiences of being a survivor.

Furthermore, marginalised LGBTIQ victims, according to Lou et al. [35], are more likely to experience harmful outcomes associated with a lack of trust in formal supports and institutions when they make their IPV experiences known. Consequently, this also increases the reliance on informal support where victims disclose their victimisation to a close family member or friend. Informal supports respond negatively with attitudes such as disbelief or blaming the victim for their victimisation based on their sexual identity. Equitable social representations are currently missing within societies, as this impacts survivorship experiences [36]. Notably, heteronormativity creates a barrier to understanding and accepting that IPV occurs outside the traditional framing (men as the only perpetrators and women as the only victims). IPV screenings, such as domestic violence and family violence screening tools, are built around the proviso that heterosexual women can only ever be victims. However, this raises questions about the legitimacy of formal support for sexually marginalised diverse individuals as it may lead to a lack of trust and adverse impacts regarding survivorship.

## 5. Discussion

Throughout the identified literature, there is a propensity to focus exclusively on coping strategies and vulnerabilities associated with being a marginalized LGBTIQ IPV victim. Foremost, the conflation of sexual violence and IPV are problematic as they have different definitions. Etienne et al. [37] on behalf of the World Health Organisation (WHO) ( p. 149) defines *sexual violence* as any unwanted sexual act to obtain a sexual act, unwanted sexual comments or advances, or acts to traffic, or otherwise directed, against any person’s sexuality using coercion, by any person regardless of their relationship to the victim, in any setting, including but not limited to home and work. Sexual violence does not need to occur exclusively in an intimate relationship. Edwards et al. [32] and Pittman et al. [33] conflate these two acts of (sexual and intimate partner) violence as mutual and only occurring within the context of an intimate relationship. Consequently, this is problematic given the articles recency and the length of time since the Etienne et al. [37] (2002) defined the act of sexual violence.

The literature focuses on the physical aspects of violence, such as being physically kicked, punched or spat on, and less attention to psychological abuse. For example, Louman et al. [35] found that for gay men who are also CALD identifying the psychological aspects of abuse were more damaging than physical violence concerning their capacity to cope and allow themselves the vulnerability to disclose their experiences. The physical aspects of violence create more long-lasting trauma, although given the hiddenness of the non-physical violence (financial, psychological, spiritual, etc.), the signs of detecting, intervening and persecuting are significantly more challenging to identify, and many will not make the connection that these forms of violence are violent at all [33]. For example, control over finances may be considered entirely normal to some. Moreover, Strasser et al. [28] found that for CALD gay men, there were significantly more significant barriers, especially at times of disclosure. These barriers manifest in systemic beliefs that men cannot be victims and can only be a perpetrator, and there is also homophobia and racism which also carry significant issues for disclosure.

The studies identified that there is also greater trust in disclosing violence to informal support, even though many rely solely on friends and family members. Those who elect to go down this path experience disbelief, blame and repeat victimisation as the individuals who witness the disclosure are more likely to reject the individual based on their homosexual identity [28]. Based on these factors, the research also demonstrates that how LGBTIQ CALD survivors of IPV survived their experience is an under-researched topic and future research on their experiences is critical. Moreover, due to this lack of research, there are significant social repercussions because many LGBTIQ minority individuals will still disclose to their families and friends over formal support, which reinforces the deep mistrust between minority communities and social institutions. In turn, this reinforces the significance of future research in this area that explores how LGBTIQ CALD IPV survivors are able to manifest resilience. 

The distrust LGBTIQ CALD people experience may be due to the inherent heterosexist nature of domestic violence screening tools, which reinforce that only women can be victims at the hands of male perpetrators with the inherent assumption of whiteness as these tools fail to consider the nuances of diverse identities [32]. Therefore, the need for tremendous respect for diversity is still needed. Consequently, little is known about how resilient marginalized LGBTIQ survivors are. The evidence supplied so far demonstrates they are in a vulnerable position and adopt various coping strategies to assist in their experiences. While Brown [38] affirms that being vulnerable and having vulnerability is the birthplace of developing resilience, little of this is reflected within the literature. Therefore, the primary author’s PhD study seeks to fill this gap by asking individuals who survived IPV and identify as marginalized LGBTIQ individuals what strategies they adopted to help them survive and manifest resilience. Additionally, it is critical for a more in-depth understanding of state and federal institutions as they play a role in allowing individuals to learn resilience, build resilience, or obtain grit/become grittier. The following section breaks down the review’s findings further by aligning with the over-arching sociological theoretical framework for a more nuanced discussion.

### 5.1. Socio-Ecological Factors and LGBTIQ Survivorship

The developmental factors that underpin socioecological theory are based on the different outcomes people experience within society and how these coalesce to form the individual’s sense of self that influences their agency and right to self-determination [39]. Within the three major themes of this study, the capacity for LGBTIQ-identifying people to develop positively and, by extension become resilient, after surviving an abusive relationship is clouded within negative coping strategies associated with their vulnerable position in any given society. Arguably, vulnerability has different interpretations based on numerous factors. Brown [39] argues that this is the birthplace of courage and meaningful change. However, the lack of clarity surrounding the vulnerable status for LGBTIQ survivors implies the proceeding definition of being harmed and increased likelihood of being attacked. The vulnerable individual’s capacity to adopt appropriate defensive measures to protect their personhood is compromised, diminished, or taken away entirely [28,32,40]. In actuality, the state is responsible for the construction, regulation and preservation of vulnerable groups. This top-down approach ranks people into privileged categories, where some people receive assistance, and others receive little to no assistance, is undoubtedly the case, as demonstrated within this study. Figure 3 highlights the various levels of development that influence a person’s manifestations of resilience or lack thereof.

### 5.2. Self and Micro Levels

The experiences of LGBTIQ individuals, in general, align with these two levels of development. These are based upon the vulnerability LGBTIQ face based on their identity notably, for victims coming to terms with their sexuality, they are more likely to experience difficulties within themself and immediate social support networks as highlighted by Luo et al. [35] and Edward et al. [32] in the result section. For example, if the individual victim is still coming to terms with their sexuality, their perpetrator may use “outing” as a form of power and control. In doing so, the victim becomes further isolated and, in turn, controlled. Moreover, the same can be said about withholding medication for HIV-related diseases. Unfortunately, these issues, which are unique to LGBTIQ people, mirror broader social attitudes such as homophobia, biphobia and transphobia, which are normalised at the individual level within their social networks.

### 5.3. Micro and Meso Levels

For marginalised LGBTIQ people who experience IPV, there are more nuanced forces which not only influence but dictate their experiences negatively. These influences stem from cultural factors where the CALD individual’s sexually diverse identity may not be as readily accepted by their immediate social supports. Moreover, societies globally dictate the acceptance, tolerance, rejection or intolerance of LGBTIQ identity, which compounds to create unique experiences based on the multiple intersections of their identity. Due to these attitudes, the LGBTIQ CALD individual may not disclose their victimised status for fear of being ostracised from their family, friends and other social supports.

### 5.4. Exo and Macro Levels

At these levels, there is an evident and apparent lack of inclusion for those who generally identify as LGBTIQ; however, the other aspects of a person’s identity also suffer from a lack of social inclusion and representation. Therefore, the impacts are more detrimental when LGBTIQ identity and CALD identity coalesce. This lack of inclusion is mirrored in policy development most heavily, as sexuality is not considered a guiding factor for different non-cis-hetero-normative outcomes—for example, the belief that once heterosexual, always heterosexual is quite evident in places such as Australia. These assumptions damage inclusive policy development and institutions having the cultural capacity to be inclusive and create safe environments for all victims. LGBTIQ CALD people’s pathologising experiences stem from a lack of appropriate cultural normalisation for their identity, and these issues require more significant social reform within Australia and in other western countries.

## 6. Limitations

This systematic literature review offered a thorough synthesis of published primary studies. However, a possible drawback is that literature such as unpublished masters and doctoral theses and grey literature have been omitted from the search. This review is further constrained as all included peer-reviewed articles were published in the United States. As such, the views expressed are limited to those within a specific socio-cultural, political and economic context.

## 7. Conclusions

This review sought to investigate how experiences of survivorship and manifestations of resilience arise within marginalised LGBTIQ CALD survivors of IPV. The gaps within the literature demonstrate the lack of clarity surrounding survivorship experiences, with all included literature focusing on coping or vulnerability. While vulnerability can be seen as a precursor for resilient outcomes, this was not clear in the included literature. Therefore, it can be inferred that the vulnerability of LGBTIQ CALD individuals can be interpreted through multiple levels of the socioecological theory. Due to Australian society and many Westernised nation states’ inability to normalise the experiences of LGBTIQ and CALD identity, there will continue to be many compounding factors that increase their vulnerability and ensure they continue to use negative coping strategies.

Further investigation into other survivorship experiences is required, with a particular emphasis on resilient outcomes. Moreover, several recommendations are based mainly on CALD and LGBTIQ identity. Particular emphasis must be placed on policy development, including all victims, irrespective of their identity. This approach must be top-down, recognising that any individual can be a victim and any individual can be a perpetrator. Normalising the experiences of all people will ensure that those who are vulnerable, diverse, and marginalised also partake in the benefits of a society that validates and advocates diversity of IPV experiences. While these are critical steps and small gestures of inclusion, they are steps that demonstrate that nation states prioritise all their citizens’ well-being and experiences. Nation states must advocate equitably for all people without prioritising the needs of one group over another; victims are victims regardless of who they are or how they identify, and inclusive policies must respect this to ensure manifestations of resilience occur—without survivors only using negative coping strategies such as drug and alcohol dependency.

## Figures and Tables

**Figure 1 ijerph-19-15843-f001:**
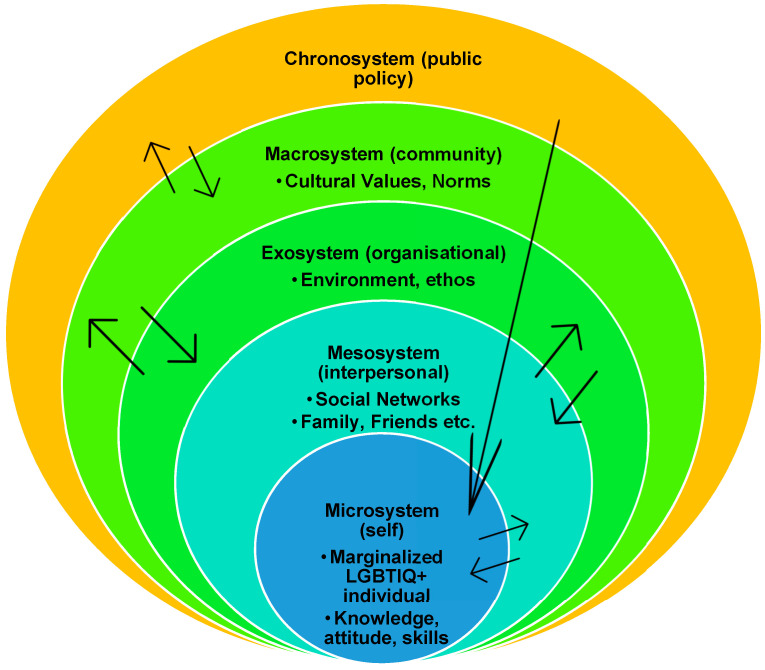
Socioecological theory, IPV and marginalised LGBTIQ people.

**Figure 2 ijerph-19-15843-f002:**
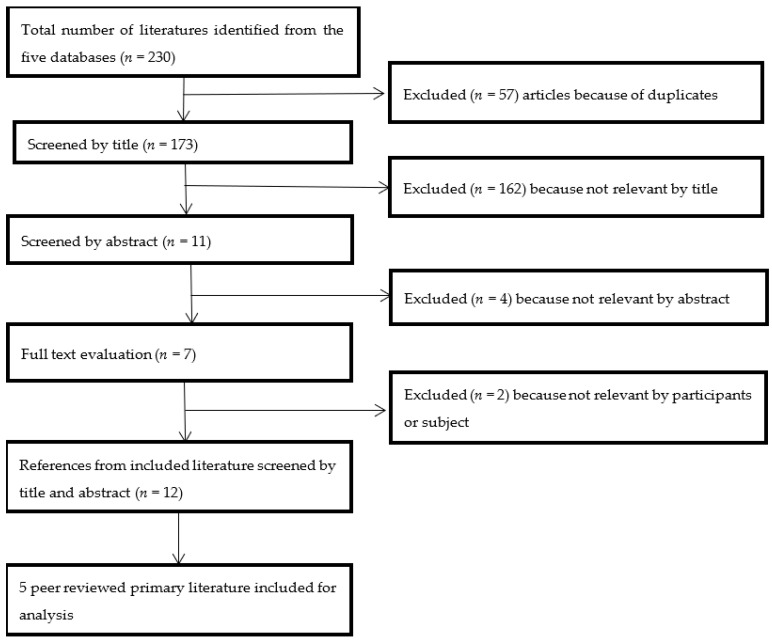
Article selection process.

**Figure 3 ijerph-19-15843-f003:**
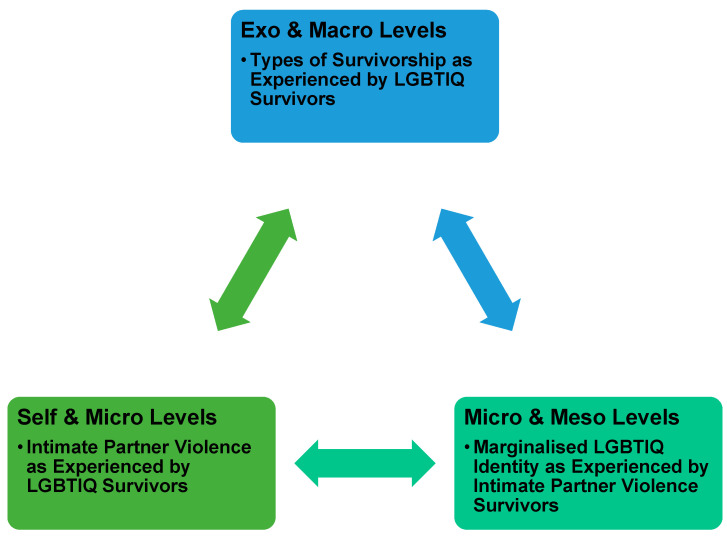
The interconnected nature of LGBTIQ IPV and Socioecological Theory.

**Table 1 ijerph-19-15843-t001:** Summary of inclusion/exclusion criteria and keywords.

Population	Inclusion	Exclusion	Keywords
Location	International	Not Applicable	Not Applicable
Language	Written in English	Other Languages	Select for English Only
Time	Not Applicable	Not Applicable	Not Applicable
Population	Publications which focus on: people from minority or marginalized populations who identify as LGBTIQ	Publications which do not focus on: People from minority or marginalized populations who identify as LGBTIQ	TITLE: (lesbian OR gay OR bisexual OR trans* OR intersex OR queer OR LGBT* OR homosexual*Or Gender* OR sexu* OR questioning OR Gender non-conforming) AND Abstract: (Visible minority OR Visual minority OR Culturally and Linguistically Diverse OR Non-White OR ethnic minority OR racial minority OR linguistic minority OR language minority OR English as Second Language OR Language other than English OR Language Background other than English OR English as an Additional Language or Dialect)
Phenomena/Target	Studies concerned with resilience and intimate partner violence	Studies not concerned with resilience and intimate partner violence	AND Abstract: (resilien* OR surviv* OR grit OR self-control OR agency OR self-sufficiency OR self-determination OR victim* Or coping OR thrive OR endur* adapt* OR fragility OR vulnera* OR weakness OR rigidity) AND TITLE: (Intimate partner violence OR partner violence OR partner abuse OR psychological abuse OR financial abuse OR physical violence OR domestic violence OR family violence)
Study/Literature Type	Peer-reviewed primary published research academic journals	Literature not included: peer-reviewed primary published research academic journals	Not Applicable

**Table 2 ijerph-19-15843-t002:** Characteristics of included studies.

No#	Author/Year	Country of Study	Sample Size	Demographics of Participants	Type of Violence	Type of Survivorship Discussed	Study Design/Data Collection Method	Data Analysis	Theoretical Approach
**1**	Edwards, Waterman, Ulman, Rodriguez, Dardis & Dworkin (2020)	USA	1268 participants	LGBT identifying white and non-white minorities, heterosexual participants	Partner violence and sexual violence	Coping	Surveys	Chi-squares and T tests	Supporting survivors and self (SSS), attribution theory and planned behaviour theory
**2**	Pittman, Ridey Rush, Hurley & Minges (2020)	USA	9435 Participants	Women of colour who identify as sexual minorities (i.e., lesbians, bisexual, etc.)	Intimate partner violence and sexual violence	Vulnerability	Self-elected National Health Assessment Data	*t*-tests	Intersectionality theory
**3**	Strasser, Smith, Pendrick-Denney, Boos-Beddington, Chen & McCarthy (2012)	USA	100 Participants	Gay and bisexual males	Intimate partner violence	Coping	Cross-sectional surveys	Chi-square tests	Does not specify
**4**	Whitton, Dyar, Mustanski & Newcomb (2019)	USA	352 Participants	Age: 16–32 Women assigned female at birth LGBT identifying. From pre-existing cohort study.	Intimate partner violence including coercive control.	Vulnerability	Pre-existing cohort study	Latent class analysis	Minority stress theory
**5**	Lou, Stone & Tharp (2014)	USA	62,861 Participants	LGBTQ (questioning) white and non-white identifying	Dating violence	Coping	Does not specify	Logistic regression	Does not specify

**Table 3 ijerph-19-15843-t003:** CASP Cohort Study.

Questions		Are the Results of the Study Valid?	Section A: Are the Results of the Study Valid?							Section B: What Are the Results?			Section C: Will the Results Help Locally?			AW	TD
**Legend**	**Y-Yes/CT-Can’t Tell/N-No**	**Y/CT/N**	**Y/CT/N**	**Y/CT/N**	**Y/CT/N**	**Y/CT/N**	**Y/CT/N**	**Y/CT/N**	**Y/CT/N**	**Y/CT/N**	**Y/CT/N**	**Y/CT/N**	**Y/CT/N**	**Y/CT/N**	**Y/CT/N**	S M W	S M W
**No. (As per Table 2)**	**Author/Year**	**Q1. Did the study address a clearly ** **focused issue?**	**Q2. Was the cohort recruited in ** ** an acceptable way?**	**Q3. Was the exposure accurately measured to minimise bias?**	**Q4. Was the outcome accurately ** ** measured to minimise bias?**	**Q5A. Have the authors identified ** ** all important confounding ** **factors?**	**Q5B. Have they take account of ** ** the confounding factors in the design and/or analysis?**	**Q6A. Was the follow up of subjects complete enough?**	**Q6B. Was the follow up of subjects long enough?**	**Q7. What are the results of this study?**	**Q8. How precise are the results?**	**Q9. Do you believe the results?**	**Q10. Can the results be applied to ** ** the local population?**	**Q11. Do the results of this study fit with other available ** ** evidence?**	**Q12. What are the implications of this study for practice?**		
**1**	Strasser et al., 2012	Y	Y	Y	Y	Y	Y	CT	CT	Y	Y	Y	Y	Y	CT	M	M
**2**	Edwards et al., 2021	Y	Y	CT	Y	Y	Y	CT	CT	Y	Y	Y	Y	Y	CT	W	W
**3**	Lou et al., 2014	Y	Y	CT	CT	Y	Y	CT	CT	Y	Y	Y	Y	Y	Y	W	W
**4**	Whitton et al., 2019	Y	CT	Y	Y	Y	Y	CT	CT	Y	Y	Y	Y	Y	Y	M	M
**5**	Pittman et al., 2020	Y	Y	CT	Y	Y	Y	CT	CT	Y	Y	Y	Y	Y	Y	M	M

## Data Availability

The data are not publicly available.

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
