# Peer review of "LGBTIQ CALD People’s Experiences of Intimate Partner Violence: A Systematic Literature Review"

_ijerph, 2022, doi:10.3390/ijerph192315843_

Round 1
Reviewer 1 Report
dear authors, I read your work and it seemed too long and dispersed. Your aim is not clear or I am not sure if you have achieved it with this systematic review of the literature. I invite you to rewrite it and make it more usable. dear authors, I read your work and it seemed excessively long and dispersed. Your goal is not clear or I am not sure if you have achieved it with this systematic review of the literature. I invite you to rewrite it and make it more understandable . furthermore I would like to point out to specify at the beginning of the text what CALD people means.Author Response
Dear reviewer.
Thank you for your comments, I have made adjustments according to your feedback. In which CALD has been addressed and defined, and the article's aim has been refined and the article has been revised with some text deleted which meets the journal requirements for standard submissions of a systematic review.
Regards
Reviewer 2 Report
Major comments:
- In the abstract, it is necessary to more clarification on lacking’s existed in the recent cited study. And what kind of extra scientific significance exists in their articles compared to existing research?
- In the introduction part, the authors need to include the most recent studies such as for the years 2022 and 2021.
- According to the research questions, the authors need to show relevance in the findings section more clearly.
- In the methods they have included only studies for the USA, Are there any specific reasons? Need to clarify it.
- It is necessary to rearrange the Discussion and conclusion part briefly with a more logical sequence according to their objective and chronological order.
- It is not clear to me how this research may be helpful for the scientific community for future research.
Minor Comments:
1. Need to update the problem of solving grammatical issues such as articles, Voice, prepositions, and some sentence patterns.
Author Response
Dear reviewer,
Thank you for your feedback and recommendations, please see below:
|
In the abstract, it is necessary to more clarification on lacking’s existed in the recent cited study. And what kind of extra scientific significance exists in their articles compared to existing research?
|
Addressed. |
|
In the introduction part, the authors need to include the most recent studies such as for the years 2022 and 2021.
|
Addressed. Please note there is not a significant body of evidence that focuses on the LGBTIQ community. |
|
According to the research questions, the authors need to show relevance in the findings section more clearly.
|
Q. 1 is addressed in theme 1: Intimate Partner Violence as Experienced by LGBTIQ Survivors Q. 2 is addressed in themes: Marginalised LGBTIQ Identity as Experienced by Intimate Partner Violence Survivors and Types of Survivorship as Experienced by LGBTIQ Survivors Q. 3 is address in theme: Types of Survivorship as Experienced by LGBTIQ Survivors A statement has been added to clarify this point.
|
|
It is necessary to rearrange the Discussion and conclusion part briefly with a more logical sequence according to their objective and chronological order.
|
A sentence has been added to address this as the reason this is broken down is due to the socioecological theory that guides this paper. Please note this is highlighted in green. |
|
In the methods they have included only studies for the USA, Are there any specific reasons? Need to clarify it.
|
This was discussed/addressed in the limitations section. |
|
It is not clear to me how this research may be helpful for the scientific community for future research.
|
Addressed in discussion and conclusion. |
|
Need to update the problem of solving grammatical issues such as articles, Voice, prepositions, and some sentence patterns. |
Addressed throughout. |
Thank you.
Round 2
Reviewer 2 Report
Thank you all authors for updating the articles according to the given comments in the first review.
Author Response
Thank you for your comments.